# Anxiety and depression among medical doctors in Catalonia, Italy, and the UK during the COVID-19 pandemic

Climent Quintana-Domeque[1,2,3,4]*, Ines Lee[5], Anwen Zhang[6], Eugenio Proto[4,6,7,8], Michele Battisti[4,6,8,9], Antonia Ho[10]

1 Department of Economics, Business School, University of Exeter, Exeter, United Kingdom, 2 GLO, Essen, Germany, 3 Department of Economics, HCEO, University of Chicago, Chicago, IL, United States of America, 4 IZA, Bonn, Germany, 5 Faculty of Economics, University of Cambridge, Cambridge, United Kingdom, 6 Department of Economics, Adam Smith Business School, University of Glasgow, Glasgow, United Kingdom, 7 CEPR, London, United Kingdom, 8 CESifo, Munich, Germany, 9 CReAM, London, United Kingdom, 10 MRC-University of Glasgow Centre for Virus Research, Glasgow, United Kingdom

* c.quintana-domeque@exeter.ac.uk

**Data Availability Statement:** Code and data files are publicly available from the Harvard Dataverse repository: https://doi.org/10.7910/DVN/DRSMYH.

## Abstract

Healthcare workers have had the longest and most direct exposure to COVID-19 and consequently may suffer from poor mental health. We conducted one of the first repeated multi-country analysis of the mental wellbeing of medical doctors (n = 5,275) at two timepoints during the COVID-19 pandemic (June 2020 and November/December 2020) to understand the prevalence of anxiety and depression, as well as associated risk factors. Rates of anxiety and depression were highest in Italy (24.6% and 20.1%, June 2020), second highest in Catalonia (15.9% and 17.4%, June 2020), and lowest in the UK (11.7% and 13.7%, June 2020). Across all countries, higher risk of anxiety and depression symptoms were found among women, individuals below 60 years old, those feeling vulnerable/exposed at work, and those reporting normal/below-normal health. We did not find systematic differences in mental health measures between the two rounds of data collection, hence we cannot discard that the mental health repercussions of the pandemic are persistent.

## Introduction

The coronavirus disease 2019 (COVID-19) pandemic has affected many individuals both directly and indirectly, disrupting routines and introducing new stressors [1]. Recent studies have shown that the pandemic has unequal effects on the psychological wellbeing of individuals, with women, younger individuals, and ethnic minorities being disproportionately affected [2, 3]. Effects also vary by occupational groups as certain jobs expose workers more directly to the disease [4–6]. The mental wellbeing of healthcare workers has been particularly affected by the pandemic [7–10].

Healthcare workers have been directly involved in the management of COVID-19 patients since the beginning of the pandemic. Drawing on the experiences of healthcare workers during the 2003 SARS outbreak, various mental health risk factors have been identified: lack of

**Funding:** The author(s) received no specific funding for this work.

**Competing interests:** The authors have declared that no competing interests exist.

personal protective equipment (PPE), overwhelming workload, lack of institutional support, and fear of infecting others [11–14]. Several studies have documented high rates of anxiety and depression symptoms among healthcare workers during the COVID-19 pandemic, with various risk factors such as fear of infection being identified as important [7, 8]. Most of the studies are from China [8, 9, 15] and a handful document similar patterns in other regions [10].

Various local and national mental health institutions around the world have begun to offer psychological assistance to those in need, with some services targeted specifically at healthcare workers [16–19]. Poor mental health among healthcare workers may have downstream effects on patients via worsened attention span, cognitive function, and clinical decision making [10]. Providing such assistance to healthcare workers requires understanding the state of their mental wellbeing, factors associated with mental health symptoms, and how these outcomes and factors vary across time and countries.

The aim of this study is to estimate the prevalence of anxiety and depression symptoms among medical doctors in multiple countries during the pandemic, as well as the risk factors associated with those symptoms. Drawing on the existing literature [7–9, 20–23], we hypothesize that certain demographic characteristics (e.g. sex and age), workplace safety (e.g. lack of necessary PPE), COVID-19 experience (e.g. directly treating COVID-19 patients), and health and lifestyle factors (e.g. long working hours) are associated with anxiety and depression.

This study provides one of the first repeated cross-country analyses of mental wellbeing among healthcare workers during the COVID-19 pandemic. Our sample comprises medical doctors working in Catalonia, Italy, or the UK in June (first data collection round) and November/December 2020 (second data collection round). Both the monthly COVID-19 prevalence and mortality rates increased between the two data collection rounds (S1 Table in S1 Appendix). COVID-19 cases per 100,000 increased from 33.3 (June 2020) to 809.6 (November 2020) in Catalonia, from 12.5 (June 2020) to 836.3 (December 2020) in Italy, and from 55.6 (June 2020) to 927.1 (November 2020) in the UK. Between the two rounds of data collection, COVID-19 deaths per 100,000 increased from 2.4 to 22.2 in Catalonia, from 2.2 to 30.7 in Italy, and from 4.4 to 17.8 in the UK.

In contrast to existing studies, our data allow us to quantify the prevalence of and risk factors associated with anxiety and depression symptoms across countries and to examine these outcomes at two timepoints. While previous studies have investigated heterogeneous samples of healthcare workers, our analysis focuses on medical doctors. The results of our study can inform how to protect and promote the mental wellbeing of medical doctors in current and future pandemics.

## Methods

We conducted an anonymous survey, The Healthcare Workers Survey, approved by the University of Exeter Business School Research Ethics Committee (eUEBS003024). Informed written consent was provided by all survey participants prior to their participation. Participants understood that they may withdraw from the study at any time. We followed the reporting guidelines of the American Association for Public Opinion Research (S2 Table in S1 Appendix).

The study is a repeated cross-sectional survey among members of 6 medical organizations: the COMB (Barcelona Medical Council) and the COMG (Girona Medical Council) in Catalonia (Spain), Anaao-Assomed (Union of physicians and healthcare executives) and the FIMMG (Union of general practitioners) in Italy, as well as the RCPSG (Royal College of Physicians and Surgeons of Glasgow) and the RCSEd (Royal College of Surgeons of Edinburgh) in the UK.

Due to different membership rules, members of the Catalan and Italian institutions work in Catalonia and Italy, while the Scottish institutions have members who work in different parts of the UK (S3 Table in S1 Appendix). The survey was designed in Qualtrics and was distributed via email by the corresponding institutions.

## Participants

Our data collection relied on the mailing lists of the respective medical organizations. The COMB invited 5,062 members in June and November 2020 (19.9%), focusing on those with medical license numbers ending in 1 or 2 (S1 Fig in S1 Appendix). This random sampling was chosen to avoid over-burdening members, given that other surveys were taking place at the same time. The other institutions sent invitations to all members. The COMG invited 3,120 members in June and November 2020. Similarly, the Anaao-Assomed invited 23,379 members, and the FIMMG invited 17,686 members in June and December 2020. The RCPSG invited 3,990 members in June 2020 and 4,300 members in November 2020 (S2 Fig in S1 Appendix). The RCSEd invited 4,992 members in June 2020 and 4,912 members in November 2020 (S3 Fig in S1 Appendix).

We included in our final sample those respondents that satisfy the following criteria (see S4 Table in S1 Appendix): (a) for whom we had information on sex, age, household composition, occupation, and specialty; (b) were working in the same region/country as the medical institution that they are a member of; (c) were medical professionals; (d) had typical work arrangements when surveyed (not retired, on leave, or shielding). In the first round, out of approximately 55,000 invited members, the final sample size was 3,025 (5.5%). In the second round, the final sample size was 2,250 (4.1%). The total number of respondents by region/country was: 1,849 in Catalonia (n = 876 in round 1, n = 973 in round 2); 2,574 in Italy (n = 1,637 in round 1, n = 937 in round 2); 852 in the UK (n = 512 in round 1, n = 340 in round 2). (S5 Table in S1 Appendix) documents the response rates across institutions.

Due to regional/country differences, including language differences (Catalan, English, Italian), the medical doctors at each institution have different occupation titles. For example, in the UK the occupational categories were: Consultant, Specialty Doctor and Associate Specialist (SAS), Specialty registrar, Junior doctor core training, Junior doctor foundation year, General practitioner, General practitioner trainee. In our statistical analysis, we accounted for both country and institutional differences in occupational titles.

## Outcomes and covariates

The outcomes of the study are anxiety and depression symptoms. Anxiety is measured with the Generalized Anxiety Disorder (GAD-7) questionnaire, a seven-item self-report anxiety questionnaire designed to assess health status during the previous two weeks [24, 25]. It has been validated as an anxiety screening tool and severity measure in different populations [24, 26–28]. GAD-7 scores range from 0 to 21. When used as a binary anxiety indicator, a score of 10 is the recommended threshold for referral for further evaluation [24, 29]. Using this threshold, the GAD-7 has sensitivity of 89% and specificity of 82% for generalized anxiety disorder [24, 29].

Depression is measured with the depression module of the Patient Health Questionnaire (PHQ-9), which focuses on the nine diagnostic criteria for DSM-IV depressive disorders [30]. It is a useful tool to assist clinicians in diagnosing depression and a reliable and valid measure of depression severity [30, 31]. It has been validated in a variety of populations [25, 32, 33]. PHQ-9 scores range from 0 to 27. When used as a binary depression indicator, 10 is the

recommended cut-off point. Using this threshold, the PHQ-9 has sensitivity and specificity of 88% for major depression [30, 32].

The following covariates were included in our analysis: demographic characteristics (sex, age, household composition), survey round (June vs. November/December 2020), perceptions about workplace safety (availability of PPE, reported feelings of vulnerability and exposure, perceived workplace concerns about workers safety), COVID-19 exposure (symptoms, directly treating COVID-19 patients, helping with COVID-19 tasks, healthcare worker deaths due to COVID-19 in the workplace), and health and lifestyle factors (self-reported health status, underlying health condition, working over 40 hours in the previous week, smoking behavior, whether the respondent had a flu vaccine this season). We created a binary variable "normal/below-normal health" that equals 1 if the respondent reported 3 or below on the 1–5 Likert scale for health status, where higher values correspond to better health, and zero otherwise. (S6 Table in S1 Appendix) describes all the variables. The replication data and code are available in S1 File.

## Statistical analysis

First, we described the demographic characteristics of our respondents. Second, we calculated the prevalence of anxiety and depression symptoms by country over time. Third, we calculated the prevalence of anxiety and depression by sex and age in each country. Fourth, we estimated the perceptions of workplace safety and exposure to COVID-19 by country over time. Finally, we used multivariable logistic regression to estimate odds ratios (ORs) for the association between anxiety (and depression) symptoms and the aforementioned covariates, controlling for occupational indicators and institutional indicators (i.e. COMB, COMG, Anaao-Assomed, FIMMG, RCPSG, RCSEd). The inclusion of both country-specific occupational indicators and institutional indicators allows us to account for region/country and institutional differences in occupational titles. Stata statistical software version 16.1 (StataCorp) was used for statistical analyses. P-values were 2-sided and statistical significance was set at p<0.05. Data were analyzed from March 4 to June 4, 2021.

## Results

Table 1 presents demographic characteristics of the participants by region/country and round. The percentage of respondents who were women and men differed across countries. In Italy, it was similar (50%). However, over 64% of respondents were women in Catalonia and below 35% were women in the UK. The age distribution of respondents also varied by country. The percentage of respondents younger than 60 years was over 83% in the UK, below 73% in Catalonia and below 60% in Italy.

**Table 1. Demographic characteristics of respondents.**

|  | Catalonia | | Italy | | UK | |
|---|---|---|---|---|---|---|
|  | **Round 1** | **Round 2** | **Round 1** | **Round 2** | **Round 1** | **Round 2** |
| Men | 311 (35.50) | 326 (33.50) | 846 (51.68) | 468 (49.95) | 358 (69.92) | 358 (65.00) |
| Women | 565 (64.50) | 647 (66.50) | 791 (48.32) | 469 (50.05) | 154 (30.08) | 154 (35.00) |
| Age < 60 | 633 (72.26) | 702 (72.15) | 907 (55.41) | 562 (59.98) | 427 (83.40) | 427 (84.12) |
| Age ≥ 60 | 243 (27.74) | 271 (27.85) | 730 (44.59) | 375 (40.02) | 85 (16.60) | 85 (15.88) |

Notes: Percentages reported in parentheses. Round 1 corresponds to June 2020 for all countries; Round 2 corresponds to November 2020 for Catalonia and the UK and December 2020 for Italy.

**Fig 1** documents the prevalence of moderate/above-moderate symptoms of anxiety (GAD-7≥10) and depression (PHQ-9≥10) by country over time. In June 2020 the prevalence of moderate anxiety (panel A) was higher in Italy (24.6% [95% CI, 22.5%-26.7%]) than in Catalonia (15.9% [95% CI, 13.4%-18.3%]) and the UK (11.7% [95% CI, 8.9%-14.5%]). A similar conclusion emerges when looking at round 2. In June 2020, the prevalence of moderate/above-

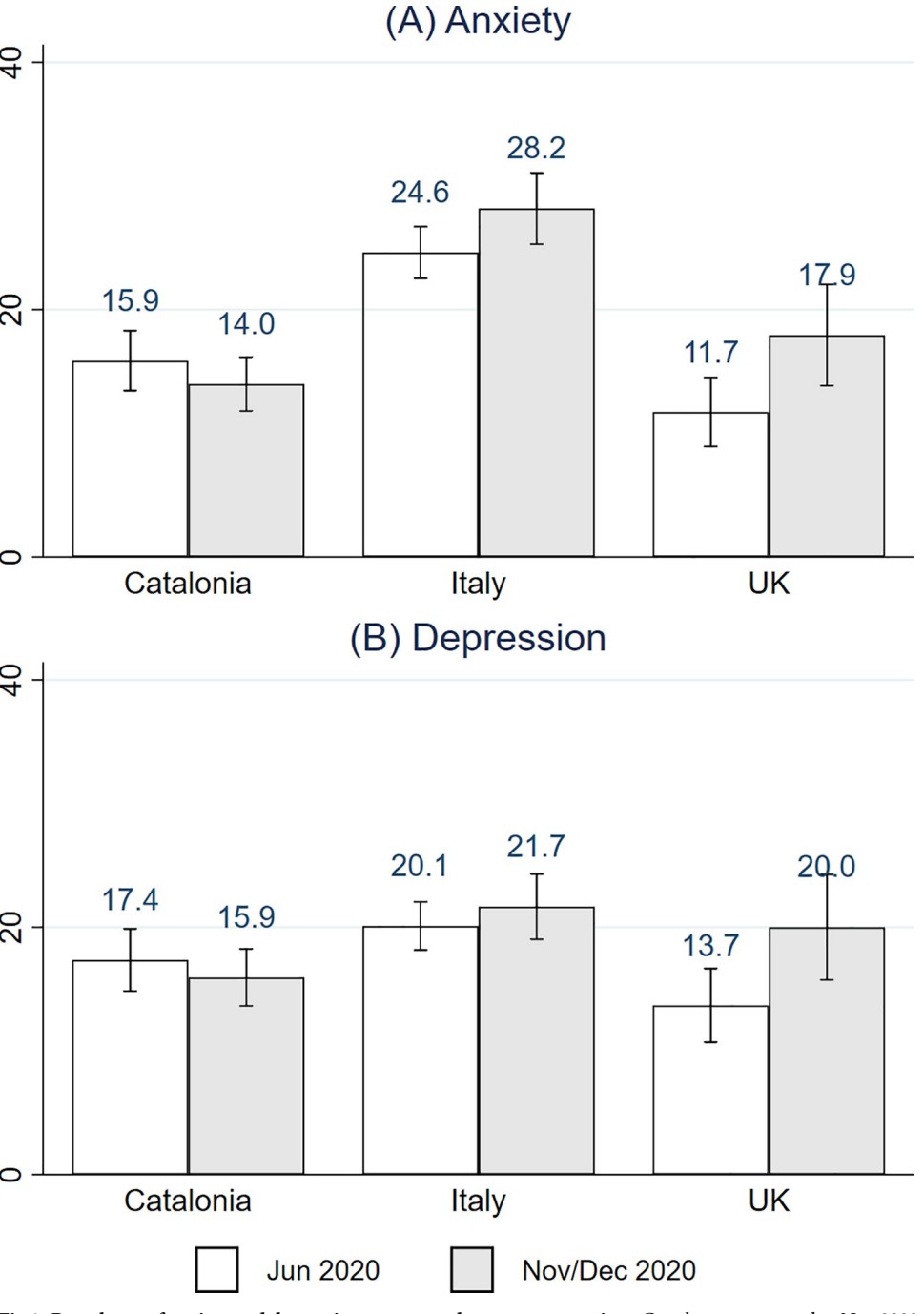

**Fig 1. Prevalence of anxiety and depression symptoms by country over time.** Grey bars correspond to Nov 2020 for Catalonia and UK, and to Dec 2020 for Italy. Anxiety symptoms = 1 if GAD-7 ≥ 10 and depression symptoms = 1 if PHQ-9 ≥ 10. 95% confidence intervals.

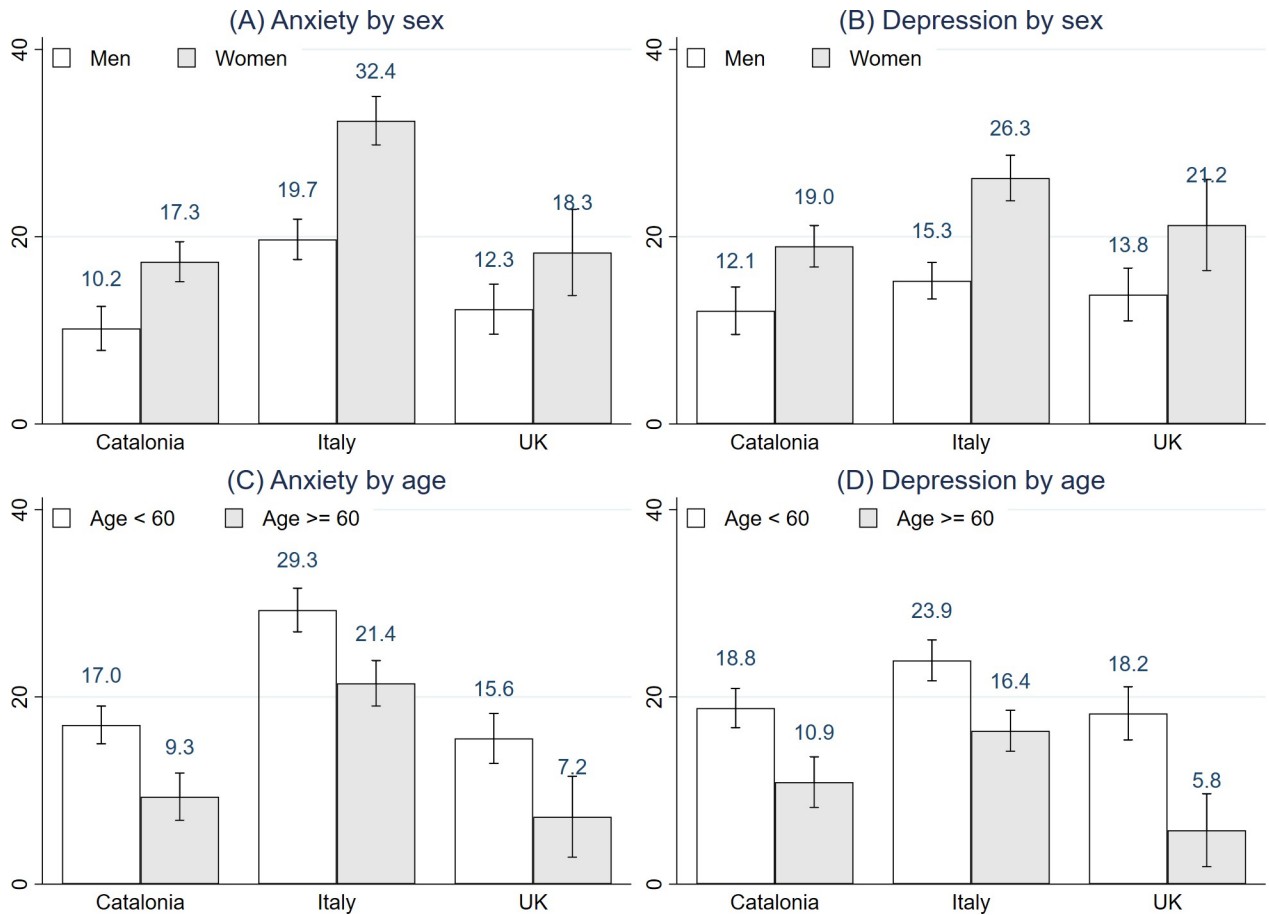

**Fig 2. Prevalence of anxiety and depression by sex and age across countries.** Grey bars are for women (panel A and B) and over 60 (panels C and B). Anxiety symptoms = 1 if GAD-7 ≥ 10 and depression symptoms = 1 if PHQ-9 ≥ 10. 95% confidence intervals.

moderate depression (panel B) was highest in Italy (20.1% [95% CI, 18.2%-22.0%]), second highest in Catalonia (17.4% [95% CI, 14.8%-19.9%]), and lowest in the UK (13.7% [95% CI, 10.7%-16.7%]). In round 2, the prevalence of moderate/above-moderate depression was higher in Italy (21.7% [95% CI, 19.0%-24.3%]) than in Catalonia (15.9% [95% CI, 13.6%-18.2%]). There were no differences in the prevalence of anxiety and depression across the two rounds of the survey in any of the countries/regions, suggesting that the mental health repercussions of the pandemic might be persistent. (S4 Fig in S1 Appendix) presents a further breakdown of the prevalence of symptoms depending on intensity.

**Fig 2** investigates the differences in the prevalence of moderate/above-moderate symptoms of anxiety and depression by sex and age across countries. Panel A shows that the prevalence of moderate/above-moderate symptoms of anxiety was higher among women than among men in Catalonia (women: 17.3%, [95% CI, 15.2%-19.5%], men: 10.2%, [95% CI, 7.9%-12.6%]) and Italy (women: 32.4%, [95% CI, 29.8%-35.0%], men: 19.7%, [95% CI, 17.6%-21.9%]). Panel B shows that the prevalence of moderate/above-moderate symptoms of depression was higher among women than among men in Catalonia (women: 19.0%, [95% CI, 16.8%-21.2%], men: 12.1%, [95% CI, 9.6%-14.6%]) and Italy (women: 26.3%, [95% CI, 23.8%-28.7%], men: 15.3%, [95% CI, 13.4%-17.3%]). Panel C shows that the prevalence of moderate/above-moderate symptoms of anxiety was higher among younger (<60 y) than older (≥60 y) respondents in all

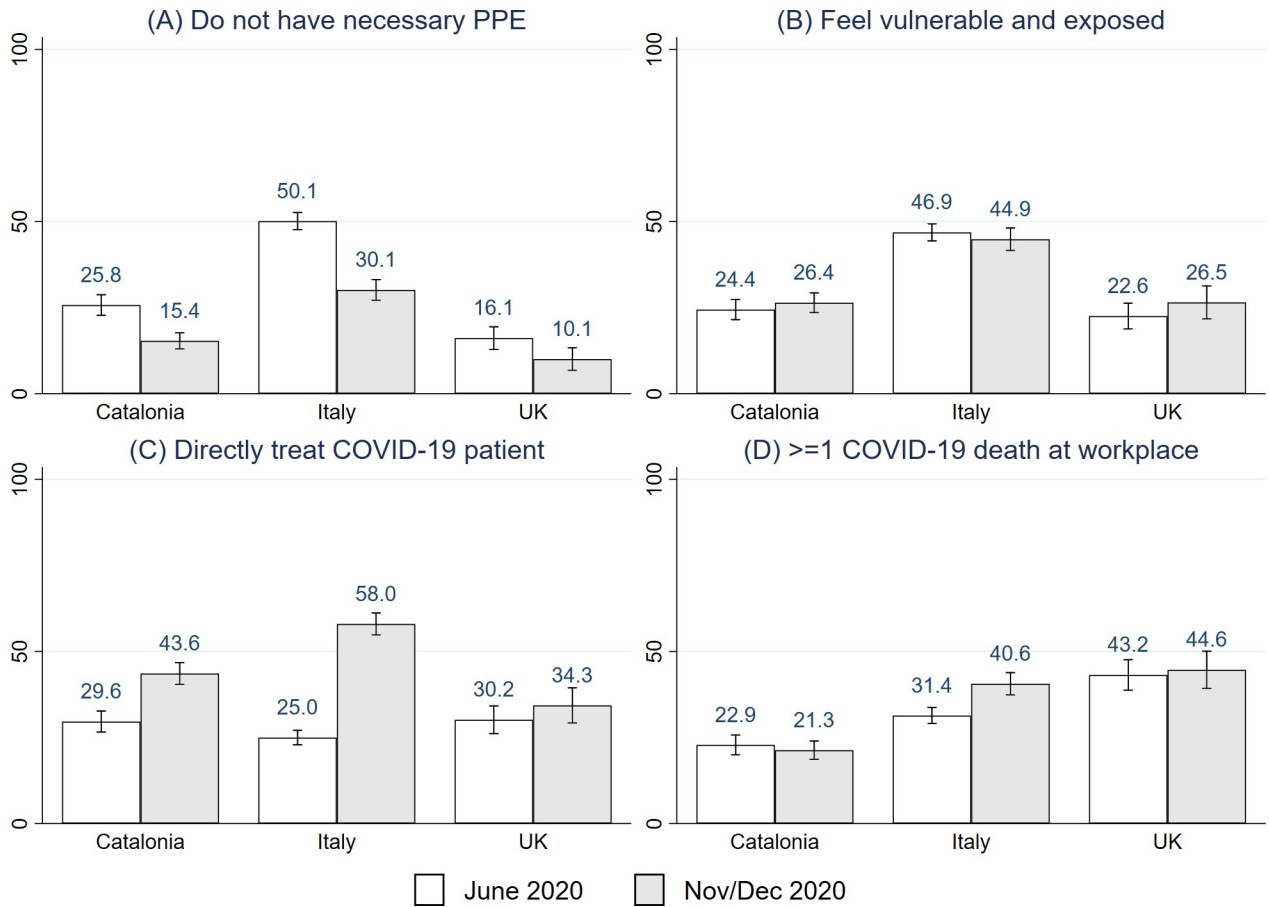

**Fig 3. Perceptions of safety and COVID-19 exposure in the workplace.** Panel A: Percentage of respondents who did not agree with the statement "my workplace is providing me with the necessary Protective Personal Equipment". Panel B: Percentage of respondents who agreed/strongly agreed with the statement "I feel vulnerable and exposed at work". Panel C: Percentage of respondents who "directly looked after COVID-19 patients last week". Panel D: Percentage of respondents who were aware of "at least one COVID-19 death among healthcare workers in their workplace".

three countries: Catalonia (<60 y: 17.0%, [95% CI, 15.0%-19.0%], ≥60 y: 9.3%, [95% CI, 6.8%-11.9%]), Italy (<60 y: 29.3%, [95% CI, 26.9%-31.6%], ≥60 y: 21.5%, [95% CI, 19.0%-23.9%]), and the UK (<60 y: 15.6%, [95% CI, 12.9%-18.2%], ≥60 y: 7.2%, [95% CI, 2.8%-11.5%]). Similarly, Panel D shows that the prevalence of moderate/above-moderate symptoms of depression was higher among younger (<60 y) than older (≥60 y) respondents across all countries.

Fig 3 describes perceptions of workplace safety and exposure to COVID-19. Around half of Italian respondents did not agree with the statement "my workplace is providing me with the necessary PPE" in June 2020 (50.1%, [95% CI, 47.6%-52.6%] (panel A). This decreased to 30.1% [95% CI, 27.1%-33.1%] in December 2020. In Catalonia, the percentage was 25.8% [95% CI, 22.8%-28.7%] in June 2020, and decreased to 15.4% [95% CI, 13.0%-17.7%] in November 2020. In the UK, 16.1% [95% CI, 12.9%-19.4%] of respondents disagreed with this statement in June 2020 and only 1 in 10 respondents disagreed with this statement in November 2020 (10.1%, [95% CI, 6.8%-13.3%]). Panel B shows that the percentage of respondents who agreed with the statement "I feel vulnerable and exposed at work" remained constant between rounds; including 1 in 5 respondents in Catalonia and the UK, and nearly 1 in 2 in Italy. It is notable that the country with the lowest rates of perceived workplace safety (Italy) also has the highest rates of anxiety symptoms.

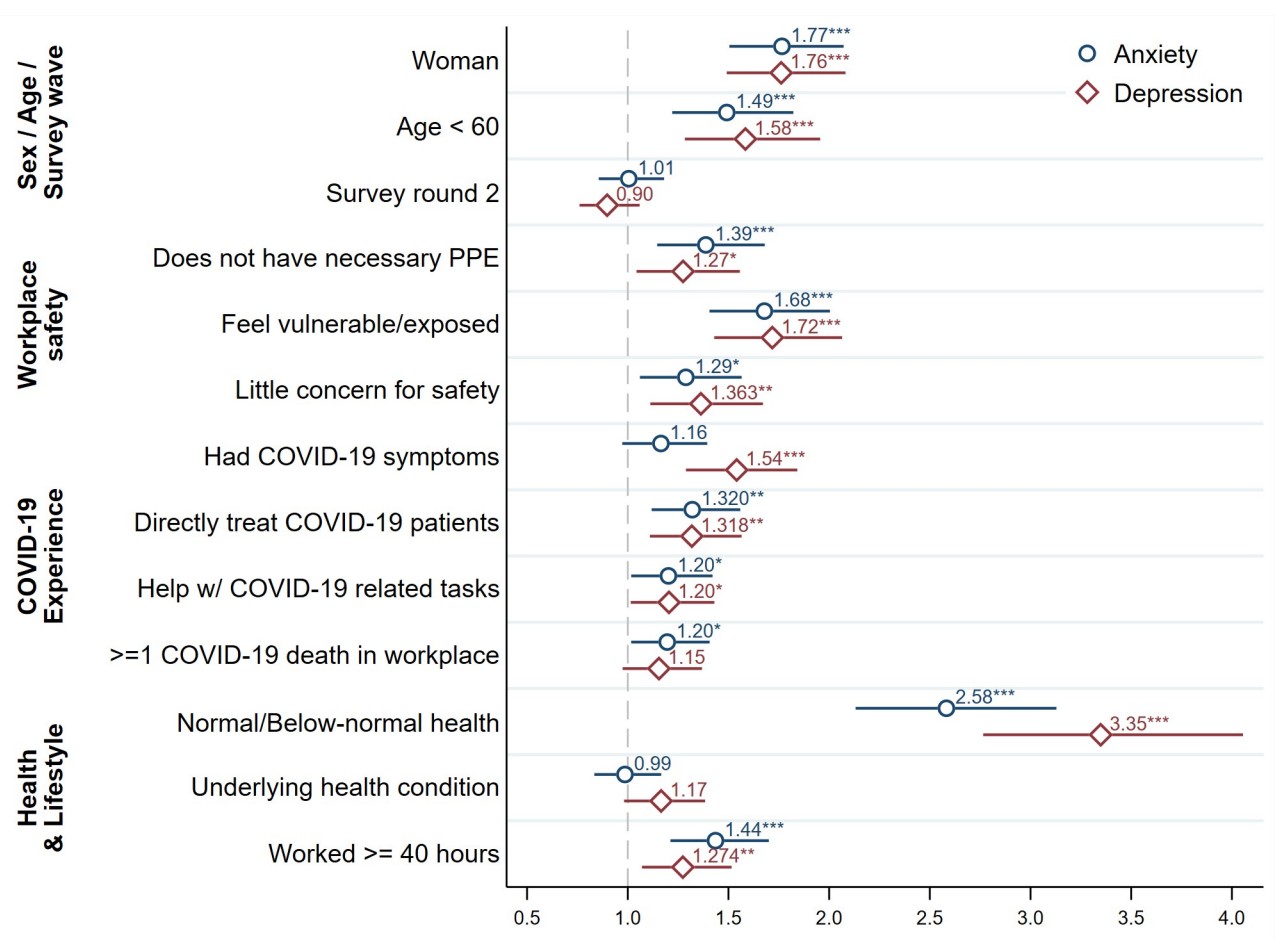

**Fig 4. Risk factors of anxiety and depression symptoms, odds ratios and 95% CI.** The figure displays the odds ratio of each variable (and its corresponding 95% CI) from a multivariable logistic regression pooling across all countries and timepoints. The dependent variables are binary indicators for anxiety (circle markers) or depression (diamond markers).

Panel C illustrates that the percentage of respondents that have "directly looked after COVID-19 patients last week" increased between June and November 2020 in Catalonia (29.6%, [95% CI, 26.6%-32.7%] to 43.4%, [95% CI, 40.4%-46.8%]) and between June and November 2020 in Italy (25.0% [95% CI, 22.9%-27.1%] to 58.0% [95% CI, 54.8%-61.2%]). In contrast, the percentage remained similar between the two rounds in the UK (30.2% [95% CI, 26.1%-34.2%] vs. 34.3% [95% CI, 29.2%-39.4%]). Lastly, Panel D shows that 1 in 5 respondents in Catalonia were aware of at least one COVID-19 death among healthcare workers in their workplace in June 2020 and November 2020. In Italy, this ratio increased from 1 in 3 in June 2020 (31.4%, [95% CI, 29.1%-33.7%]) to 2 in 5 in December 2020 (40.6%, [95% CI, 37.4%-43.8%]). In the UK, it remained constant at about 1 in 3 respondents. The increase across survey rounds in the percentage of medical doctors directly treating COVID-19 patients in the last week in Catalonia and Italy matches the evolution of the pandemic reported in S1 Table in S1 Appendix.

After pooling together all countries and the two rounds of data, **Fig 4** displays the odds-ratios (ORs) of various risk factors estimated using a multivariable logit specification with binary anxiety and depression indicators as the dependent variables. In this figure, we focus on factors that related literature has demonstrated to be correlated with mental health [7–9, 20–

23]. In addition to the variables listed in the figure, the regression also controls for indicators for smoking, a flu vaccine this season, living with a child under 5, living with someone over 60, occupational indicators, and institutional indicators. (S7 Table in S1 Appendix) reports full regression tables for the pooled sample and separate countries.

Controlling for health behaviors, household composition, occupational indicators and institutional indicators, women, younger individuals (< 60 years), those who feel vulnerable and exposed at work, those who think that their workplace has shown little concern for their safety, those who directly looked after COVID-19 patients last week, those with normal/below-normal health status (i.e. with a self-reported health status of 3 or below on a 1–5 Likert scale, where higher values correspond to better health), and those who worked over 40 hours last week had higher odds of anxiety and/or depression symptoms.

Women had higher odds of anxiety (OR = 1.77 [95% CI, 1.50–2.07]) and depression (OR = 1.76 [95% CI, 1.49–2.09]) compared to men. Compared with individuals above 60 years, younger individuals had higher odds of anxiety (OR = 1.49 [95% CI, 1.22–1.82]) and depression (OR = 1.58 [95% CI, 1.28–1.96]). A reported lack of necessary PPE in the workplace was associated with higher odds of anxiety (OR = 1.39 [95% CI, 1.15–1.68]) and depression (OR = 1.27 [95% CI, 1.04–1.56]). Respondents who felt vulnerable or exposed in their work-place had greater odds of anxiety and depression symptoms compared with respondents who did not feel vulnerable or exposed (OR = 1.68 [95% CI, 1.41–2.00]; OR = 1.72 [95% CI, 1.43–2.06]). Compared with respondents who reported above-normal health (i.e. above 3 on a 1–5 scale), individuals who reported a normal/below-normal health status (i.e. 3 or below on a 1–5 scale), had higher odds of anxiety (OR = 2.58 [95% CI, 2.13–3.13]) and depression (OR = 3.35 [95% CI, 2.76–4.06]). Compared with individuals who worked below 40 hours, those who worked 40 hours or more last week had higher odds of anxiety (OR = 1.44 [95% CI, 1.21–1.70]) and depression (OR = 1.27 [95% CI, 1.07–1.52]). (S6 Table in S1 Appendix) shows that these patterns hold for specific countries too.

## Discussion

This is one of the few studies to provide a multi-country analysis of the mental wellbeing of medical doctors at two timepoints during the COVID-19 pandemic. Among respondents from Catalonia, Italy and the UK, the prevalence of anxiety and depression was highest among medical doctors in Italy, with 1 in 4 suffering from anxiety symptoms in June and December 2020 and 1 in 5 suffering from depression symptoms over the same period. Within each country, no difference in the prevalence of anxiety and depression were reported between the first and second rounds of the survey. Hence, we cannot discard that the mental health repercussions of the pandemic are persistent.

In Catalonia, Italy, and the UK, higher risk of anxiety and depression symptoms was found among women, individuals below 60 years old, those feeling vulnerable/exposed at work, and those reporting normal/below-normal health. These associated risk factors provide a few possible reasons for the variation in the prevalence of anxiety and depression across countries. For example, the percentage of respondents who reported a lack of necessary PPE and reported feeling vulnerable and exposed at work was highest in Italy, where rates of anxiety and depression were also highest.

Our findings are consistent with other studies that have examined the rates of anxiety and depression among healthcare workers during the pandemic. In Spain, Alonso et al. [20] find that among 9,138 respondents (26.4% physicians, 77.3% women) working in 6 places (including Catalonia) during May and September 2020, the prevalence of depression (PHQ-8≥10) and anxiety (GAD-7≥10) was 22.4% (vs. 16.5% in our Catalan sample) and 17.0% (vs. 14.9%

in our Catalan sample). In Italy, Rossi et al. [21] find that among 1,379 respondents (31.4% physicians, 77.2% women) surveyed in March 2020, the prevalence of depression (PHQ-9≥15) and anxiety (GAD-7≥15) was 24.7% (vs. 6.6% in our Italian sample) and 19.8% (vs. 9.5% in our Italian sample). Conti et al. [34, 35] find similar magnitudes. In the UK, Greenberg et al. [36] find that among 709 staff (41% doctors) working in English hospitals in summer 2020, the prevalence of moderate (PHQ-9≥10) and severe (PHQ-9≥20) depression was 37% and 6%, respectively (26% and 6% among doctors, vs. 16.2% and 2.7% in our UK sample), and that the prevalence of moderate (GAD-7≥10) and severe (GAD-7≥15) anxiety was 27% and 11%, respectively (20% and 8% among doctors, vs. 14.2% and 5.9% in our UK sample). Greene et al. [37] find comparable rates.

Our findings are also consistent with studies investigating risk factors of mental health among healthcare workers. In a review of 24 studies, De Kock et al. [7] show that risk factors included underlying health, being female, concerns about workplace safety [22, 23], contact with COVID-19 [8, 38, 39], and concerns about the wellbeing of others [22]. In Spain, Alonso et al. [20] find that healthcare professionals frequently exposed to COVID-19 patients were statistically significantly more likely to experience mental health disorders (OR = 3.98, 95% CI: 3.27–4.85). In Italy, Rossi et al. [21] find that being female was associated with higher GAD-7 (OR = 2.18, 95% CI: 1.49–3.19) and PHQ-9 scores (OR = 2.03, 95% CI: 1.44–2.87). In the UK, Siddiqui et al. [40] find that among 558 healthcare professionals (51% doctors, 31% nurses), concerns about exposure to COVID-19 and the lack of PPE were important causes of anxiety.

## Contributions

Our study contributes to monitoring the mental wellbeing of medical doctors during the COVID-19 pandemic. Including multiple countries and timepoints allow comparison between different settings, and improves our understanding about how medical doctors have been affected at different points during the pandemic. The similar patterns across countries suggest that our findings may be generalizable to other European settings. Rather than relying on online convenience samples, our sampling technique relies on the institutional mailing lists of medical organizations. In comparison to previous studies, we focus on medical doctors rather than a broader group of healthcare workers.

## Limitations

This study has several limitations. First, when comparing two cross-sectional surveys for each country, we were not necessarily comparing the same individuals. Differences in prevalence of mental health symptoms could be driven by changes in sample composition across waves. Relatedly, the survey did not take place at the same point during an epidemic wave in data collection rounds 1 and 2.

Second, participants in our survey are not necessarily representative of the underlying populations of medical doctors and may be self-selected since they voluntarily take part in the survey. Reporting bias is likely. If individuals with symptoms were more likely to respond (e.g. to express grievances), then our estimates may be higher than the population average. Conversely, if individuals with above-average symptoms were less likely to respond (e.g. due to time constraints), then our estimates may be below the population average.

Third, anxiety and depression symptoms may not be comparable across countries due to different reporting norms in the GAD-7 and PHQ-9 questionnaires. Reassuringly, our pooled multivariable logistic regressions produce similar results even when controlling for occupation and institution indicators.

Finally, our measures of anxiety and depression are not based on an objective diagnosis made by a clinician.

## Conclusion

The COVID-19 pandemic has been classified as a traumatic event [1]. Healthcare workers have arguably had the most direct and longest exposure to this disease. Our study identified a high prevalence of anxiety and depression among medical doctors in both the first and second waves of the pandemic, contributing to a wider literature examining the effects of traumatic events on mental wellbeing [41], especially on those who are most exposed because of the demands of their occupation. The results of this study suggest that institutional support for healthcare workers, and in particular medical workers, is important in protecting and promoting their mental health in the current and in future pandemics.

## Supporting information

**S1 Appendix. Online supporting materials.**
(PDF)

**S1 File. Files (code and data) to replicate the findings in this article.**
(7Z)

## Acknowledgments

We are grateful to Fundació Galatea, COMB (Barcelona Medical Council), COMG (Girona Medical Council), Anaao-Assomed (Union of physicians and healthcare executives), FIMMG (Union of general practitioners), RCPSG (Royal College of Physicians and Surgeons of Glasgow), and RCSEd (Royal College of Surgeons of Edinburgh) for their collaboration. We are extremely grateful to the participants in our survey. We thank Sr. Miquel Creixell Tramuns and Dr. Climent Quintana Escandell for their help in the design and translation of the survey in Catalan, Dr. Roberto Rotundo for their help in the design and translation of the survey in Italian, and Sra. Anna Mitjans, Sra. Mariajosé Verdú, Dr. Jaume Padrós i Selma, Dr. Josep Vilaplana, Prof. Jackie Taylor, Mr. David Thomson, Dottoresa Silvia Procaccini, Prof. Paolo Misericordia, Prof. Anton Muscatelli, Prof. Vincenzo Galasso and Dr. Francesco Longo for their help in distributing our survey. Finally, we thank one anonymous referee for their valuable feedback. None of the cited papers in this article have been retracted as shown by https://retractionwatch.com as of 31st August 2021.

## Author Contributions

**Conceptualization:** Climent Quintana-Domeque, Ines Lee, Anwen Zhang, Eugenio Proto, Michele Battisti, Antonia Ho.

**Data curation:** Climent Quintana-Domeque, Ines Lee, Anwen Zhang.

**Formal analysis:** Climent Quintana-Domeque, Ines Lee, Anwen Zhang, Eugenio Proto, Michele Battisti.

**Investigation:** Climent Quintana-Domeque, Ines Lee, Anwen Zhang, Eugenio Proto, Michele Battisti, Antonia Ho.

**Methodology:** Climent Quintana-Domeque, Ines Lee, Anwen Zhang, Eugenio Proto, Michele Battisti.

**Project administration:** Climent Quintana-Domeque.

**Resources:** Climent Quintana-Domeque, Antonia Ho.

**Software:** Climent Quintana-Domeque, Ines Lee.

**Supervision:** Climent Quintana-Domeque.

**Validation:** Climent Quintana-Domeque, Ines Lee.

**Visualization:** Climent Quintana-Domeque, Ines Lee.

**Writing – original draft:** Climent Quintana-Domeque, Ines Lee, Anwen Zhang, Eugenio Proto.

**Writing – review & editing:** Climent Quintana-Domeque, Ines Lee, Anwen Zhang, Eugenio Proto, Michele Battisti, Antonia Ho.

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
