## [Decision Letter · Decision Letter 0]

24 Aug 2021

PONE-D-21-22042

Anxiety and Depression among Medical Doctors in Catalonia, Italy, and the UK during the COVID-19 Pandemic

PLOS ONE

Dear Dr. Quintana-Domeque,

Thank you for submitting your manuscript to PLOS ONE. After careful consideration, we feel that it has merit but does not fully meet PLOS ONE’s publication criteria as it currently stands. Therefore, we invite you to submit a revised version of the manuscript that addresses the points raised during the review process.

We look forward to receiving your revised manuscript.

Kind regards,

Gabriel A. Picone

Academic Editor

PLOS ONE

Journal Requirements:

Reviewers' comments:

Reviewer's Responses to Questions

**Comments to the Author**

1. Is the manuscript technically sound, and do the data support the conclusions?

Reviewer #1: Yes

2. Has the statistical analysis been performed appropriately and rigorously? 

Reviewer #1: Yes

3. Have the authors made all data underlying the findings in their manuscript fully available?

Reviewer #1: Yes

4. Is the manuscript presented in an intelligible fashion and written in standard English?

Reviewer #1: Yes

5. Review Comments to the Author

Reviewer #1: I am pleased to send my comments for the Manuscript “Anxiety and Depression among Medical Doctors in Catalonia, Italy, and the UK during the COVID-19 Pandemic”. This manuscript describes a cross-country study aimed to investigate the prevalence of anxiety and depression in medical doctors at two-time points during the COVID-19 pandemic. The Authors address an important topic, the manuscript is a valuable contribution to the literature and is written clearly. I would ask the Authors to address minor amendments, as follows:

1. In the Introduction section, the Authors should specify the aims and the hypotheses of the study.

2. In the Methods section:

- The inclusion and exclusion criteria of the sample should be specified.

- In the Participants section, I suggest describing the occupational categories of participants. For example, in the Statistical Analysis section, the Authors reported the following occupational categories for UK participants (line 102, p. 6): “Consultant, SAS doctor, Specialty registrar, Junior doctor core training, Junior doctor foundation year, General practitioner, General practitioner trainee”. To avoid repetitions, this sentence could be moved and extended in the description of the sample.

3. In the Results section:

- To avoid reader confusion, the sentence on lines 109-112 (p. 6) should be moved to the Participants section. In fact, this sentence refers to the original and final sample sizes and not to Table 1 of the Results.

- The sentence on lines 116-121 (pp. 6 - 7) describes the increase in COVID-19 cases during the data collection period. It is a description of the context in which the study was carried out, rather than a result of the Authors. For this reason, I suggest moving this sentence to the Introduction section. Panel B could be moved to supplementary material.

- I recommend summarizing the description of Figure 4 (lines 199-205, p. 10) and reporting only the essential information. The sentence at lines 202-205 [“In addition to the variable listed in the figure, the regression also controls for indicators for smoking, a flu vaccine this season, living with a child under 5, living with someone over 60, occupational codes, and institutional codes. Table S6 (S1 Appendix) reports full regression tables for the pooled sample and separate countries”] could be eliminated.

- Referring to logistic regression analysis, it is unclear why the Authors did not show results for some covariates (i.e., health behaviors, household composition, occupational categories, and medical organization indicators). This should be clarified, and the results described.

4. In the Discussions section, the Authors state that “Within each country, no difference in the prevalence of anxiety and depression were reported between the first and second rounds of the survey” (lines 224 – 226, p. 11). However, this is not specified in the Results section. I recommend describing these results also in the Results section, referring to the table where the data are reported.

6. PLOS authors have the option to publish the peer review history of their article (what does this mean?). If published, this will include your full peer review and any attached files.

Reviewer #1: No

---

## [Author Response · Author response to Decision Letter 0]

7 Sep 2021

[1.] In the Introduction section, the Authors should specify the aims and the hypotheses of the study.

Answer: Thank you for this suggestion. We have now specified the aims and hypotheses of the study in Paragraph 4 of the introduction. Specifically, our aim is to estimate the prevalence of anxiety and depression symptoms among medical doctors in multiple countries during the pandemic, as well as the risk factors associated with those symptoms. Drawing on the existing literature, our hypothesis is that certain demographic characteristics (e.g. sex and age), workplace safety (e.g. lack of necessary PPE), COVID-19 experience (e.g. directly treating COVID-19 patients), and health and lifestyle factors (e.g. long working hours) are associated with anxiety and depression. 

[2.] In the Methods section:

The inclusion and exclusion criteria of the sample should be specified. 

Answer: Thank you for this suggestion. Paragraph 2 of the “Participants” section specifies the criteria for including respondents in our sample. 

In the Participants section, I suggest describing the occupational categories of participants. For example, in the Statistical Analysis section, the Authors reported the following occupational categories for UK participants (line 102, p. 6): “Consultant, SAS doctor, Specialty registrar, Junior doctor core training, Junior doctor foundation year, General practitioner, General practitioner trainee”. To avoid repetitions, this sentence could be moved and extended in the description of the sample.

Answer: In Paragraph 3 of the “Participants” section, we clarify that due to regional/country differences, including language differences (English, Catalan, Italian), the medical doctors at each institution have different occupation titles. We provide the UK occupation category as an example to avoid translation-related issues. In this paragraph, we also clarify that we account for institutional differences in occupational titles in our statistical analysis. Our methods for doing so (including indicators for all occupational categories and institutions) is expanded further in the “Statistical analysis section”. 

[3.] In the Results section:

To avoid reader confusion, the sentence on lines 109-112 (p. 6) should be moved to the Participants section. In fact, this sentence refers to the original and final sample sizes and not to Table 1 of the Results.

Answer: Thank you for this suggestion. We have moved sentences on lines 109-112 in the original manuscript to the “Participants” section. 

The sentence on lines 116-121 (pp. 6 - 7) describes the increase in COVID-19 cases during the data collection period. It is a description of the context in which the study was carried out, rather than a result of the Authors. For this reason, I suggest moving this sentence to the Introduction section. Panel B could be moved to supplementary material.

Answer: Thank you for this suggestion. We have moved sentences on lines 116-121 in the original manuscript to the introduction. The information in Table 1, Panel B of the original manuscript is now presented in S1 Appendix, Table S1. 

I recommend summarizing the description of Figure 4 (lines 199-205, p. 10) and reporting only the essential information. The sentence at lines 202-205 [“In addition to the variable listed in the figure, the regression also controls for indicators for smoking, a flu vaccine this season, living with a child under 5, living with someone over 60, occupational codes, and institutional codes. Table S6 (S1 Appendix) reports full regression tables for the pooled sample and separate countries”] could be eliminated.

Answer: Thank you for this suggestion. We have condensed the legend of figure 4 and moved the additional detail to the main text. 

Referring to logistic regression analysis, it is unclear why the Authors did not show results for some covariates (i.e., health behaviors, household composition, occupational categories, and medical organization indicators). This should be clarified, and the results described.

Answer: Thank you for this clarification. We chose to focus on the variables in figure 4 because these are the key variables that the related literature has demonstrated to be correlated with mental health (namely demographic characteristics, workplace safety, COVID-19 experience, lifestyle factors) and therefore forms the basis of our hypotheses. In the updated manuscript, we provide an explanation for why we focus on these sets of covariates. 

[4.] In the Discussions section, the Authors state that “Within each country, no difference in the prevalence of anxiety and depression were reported between the first and second rounds of the survey” (lines 224 – 226, p. 11). However, this is not specified in the Results section. I recommend describing these results also in the Results section, referring to the table where the data are reported.

Answer: Thank you for this suggestion. We have included a sentence at the end of Paragraph 2 of the “Results” section describing the similarity of anxiety and depression rates across the two rounds of data collection.

---

## [Decision Letter · Decision Letter 1]

15 Oct 2021

Anxiety and Depression among Medical Doctors in Catalonia, Italy, and the UK during the COVID-19 Pandemic

PONE-D-21-22042R1

Dear Dr. Quintana-Domeque,

We’re pleased to inform you that your manuscript has been judged scientifically suitable for publication and will be formally accepted for publication once it meets all outstanding technical requirements.

Kind regards,

Gabriel A. Picone

Academic Editor

PLOS ONE

Additional Editor Comments (optional):

Reviewers' comments:

Reviewer's Responses to Questions

**Comments to the Author**

1. If the authors have adequately addressed your comments raised in a previous round of review and you feel that this manuscript is now acceptable for publication, you may indicate that here to bypass the “Comments to the Author” section, enter your conflict of interest statement in the “Confidential to Editor” section, and submit your "Accept" recommendation.

Reviewer #1: All comments have been addressed

2. Is the manuscript technically sound, and do the data support the conclusions?

Reviewer #1: Yes

3. Has the statistical analysis been performed appropriately and rigorously? 

Reviewer #1: Yes

4. Have the authors made all data underlying the findings in their manuscript fully available?

Reviewer #1: Yes

5. Is the manuscript presented in an intelligible fashion and written in standard English?

Reviewer #1: Yes

6. Review Comments to the Author

Reviewer #1: I believe the Authors have responded well to my comments and suggestions. Thus, I endorse the current version of the manuscript for publication.

7. PLOS authors have the option to publish the peer review history of their article (what does this mean?). If published, this will include your full peer review and any attached files.

Reviewer #1: **Yes: **Roberta Lanzara

---

## [Editor Report · Acceptance letter]

20 Oct 2021

PONE-D-21-22042R1 

Anxiety and Depression among Medical Doctors in Catalonia, Italy, and the UK during the COVID-19 Pandemic 

Dear Dr. Quintana-Domeque:

I'm pleased to inform you that your manuscript has been deemed suitable for publication in PLOS ONE. Congratulations! Your manuscript is now with our production department. 

Kind regards, 

on behalf of

Dr. Gabriel A. Picone 

Academic Editor

PLOS ONE